# Maximal Fat Oxidation during Incremental Upper and Lower Body Exercise in Healthy Young Males

**DOI:** 10.3390/ijerph192215311

**Published:** 2022-11-19

**Authors:** Mike Price, Lindsay Bottoms, Matthew Hill, Roger Eston

**Affiliations:** 1Centre for Sport, Exercise and Life Sciences, School of Life Sciences, Coventry University, Coventry CV1 5FB, UK; 2Department of Psychology, Sport and Geography, University of Hertfordshire, Hatfield AL10 9AB, UK; 3Alliance for Research in Exercise, Nutrition and Activity, Campus Central—City East, University of South Australia, GPO Box 2471, Adelaide 5001, Australia

**Keywords:** Fat_max_, carbohydrate oxidation, cycle ergometry, arm crank ergometry, variability

## Abstract

The aim of this study is to determine the magnitude of maximal fat oxidation (MFO) during incremental upper and lower body exercise. Thirteen non-specifically trained male participants (19.3 ± 0.5 y, 78.1 ± 9.1 kg body mass) volunteered for this repeated-measures study, which had received university ethics committee approval. Participants undertook two incremental arm crank (ACE) and cycle ergometry (CE) exercise tests to volitional exhaustion. The first test for each mode served as habituation. The second test was an individualised protocol, beginning at 40% of the peak power output (PO_peak_) achieved in the first test, with increases of 10% PO_peak_ until volitional exhaustion. Expired gases were recorded at the end of each incremental stage, from which fat and carbohydrate oxidation rates were calculated. MFO was taken as the greatest fat oxidation value during incremental exercise and expressed relative to peak oxygen uptake (%$\dot{V}$O_2peak_). MFO was lower during ACE (0.44 ± 0.24 g·min^−1^) than CE (0.77 ± 0.31 g·min^−1^; respectively, *p* < 0.01) and occurred at a lower exercise intensity (53 ± 21 vs. 67 ± 18%$\dot{V}$O_2peak_; respectively, *p* < 0.01). Inter-participant variability for MFO was greatest during ACE. These results suggest that weight loss programs involving the upper body should occur at lower exercise intensities than for the lower body.

## 1. Introduction

During the initial intensities of incremental exercise, both carbohydrate and fat oxidation increase [1]. However, in subsequent stages of submaximal exercise, carbohydrate oxidation continues to increase, whereas fat oxidation reaches maximal values and begins to decrease [1,2]. Within such an incremental exercise model, the greatest fat oxidation value observed is termed maximal fat oxidation (MFO). The exercise intensity at which the MFO occurs, typically a percentage of maximal or peak oxygen uptake ($\dot{V}$O_2peak_), has been termed ‘Fat_max_’ [3]. The intensity at which Fat_max_ occurs has been shown to be affected by factors such as endurance training status [4], prior carbohydrate ingestion [5], habitual physical activity and maximal oxygen uptake ($\dot{V}$O_2max_) [6]. Furthermore, few studies have directly compared males and females, with MFO tending to be greater in males but Fat_max_ tending to be greater in females [7]. Although the determination of the exercise intensity at which Fat_max_ occurs has been reported to be reliable, there is often large inter- and intra-individual variation, as demonstrated by large coefficients of variation and not all participants demonstrating a clear Fat_max_ intensity [6,8].

Previous studies examining Fat_max_ have predominantly considered lower body exercise. However, for the general population, upper body exercise elicits lower $\dot{V}$O_2peak_ than for lower body exercise modes [9,10,11,12]. For example, values for $\dot{V}$O_2peak_ during arm crank ergometry (ACE) and cycle ergometry (CE) in non-specifically trained individuals have been reported as 34 and 48 mL·kg·^−1^min.^−1^, respectively [10], with upper body values generally representing ~70% of those during CE [10,11,13]. When $\dot{V}$O_2peak_ during ACE (e.g., 28–39 mL·kg·^−1^min.^−1^) is compared to that during treadmill running (e.g., 44–59 mL·kg·^−1^min.^−1^) in similar non-specifically trained populations [9,12], the proportional difference between modes is increased, with ACE being ~60–65% of treadmill values [9,12]. Differences in $\dot{V}$O_2peak_ across exercise modes are predominantly due to differences in the size of the active muscle mass. The physiological range for metabolic values during upper body exercise is thus lower than for other exercise modes and likely includes a reduced range of fat metabolism.

Maximal fat oxidation and Fat_max_ during typical incremental CE and treadmill walking/running protocols have been directly compared in moderately trained men [14]. Maximal oxygen uptake and fat oxidation were lower during CE (64 mL·kg·^−1^min.^−1^ and 0.47 g·min^−1^, respectively) than for treadmill exercise (67 mL·kg·^−1^min.^−1,^ and 0.65 g·min^−1^, respectively), as was fat oxidation over a range of submaximal exercise intensities. However, Fat_max_ was similar between modes (~60%$\dot{V}$O_2peak_), as was the point where blood lactate increased from rest (i.e., the lactate threshold), suggesting no exercise mode differences for these specific variables. The participants in the above study were club/county standard endurance cyclists and, therefore, of greater aerobic training status than the general population, where fitness indicators such as lactate threshold are lower in ACE when compared to CE and treadmill exercise [12]. Differences in MFO and Fat_max_ across exercise modes may, therefore, be more pronounced for non-specifically trained individuals.

As well as eliciting lower aerobic fitness indicators when compared to lower body exercise modes, the upper body is also considered to possess a greater proportion of fast twitch muscle fibres [15,16]. Considering that fast twitch fibres are more glycolytic than slow twitch fibres and that lower training status elicits lower absolute MFO values at lower relative exercise intensities [4,6], the magnitude of both MFO and Fat_max_ is likely lower for upper body exercise. Studies that have investigated Fat_max_ during upper body exercise have examined individuals with diabetes and obesity, specifically focusing on mitochondrial function [17,18], or those with spinal cord injury, focusing upon substrate utilisation [19]. Within these studies, healthy or non-disabled controls have been utilised and matched for age and body mass [17,18].

Based on the above studies, our knowledge of MFO and Fat_max_ during upper body exercise is from a small number of studies of older (~30–40 years of age) and heavier participants (>90 kg) with low aerobic fitness; all factors associated with poor MFO and Fat_max_ intensities [20]. Therefore, the aim of this study is to determine the magnitude of MFO and Fat_max_ during upper body and lower body exercise in young, healthy, recreationally active males. Due to the likely differences in muscle mass, metabolic capacity and training status between the upper and lower body, it is hypothesised that MFO and Fat_max_ would be lower and more variable during upper body exercise.

## 2. Materials and Methods

### 2.1. Participants

Thirteen healthy, active males volunteered to participate in this study (age; 19.3 ± 0.5 y, body mass; 78.1 ± 9.1 kg; 4.5 ± 1.8 sessions per week in predominantly team sports), which had received university ethics committee approval (Ref: P35579) and was in accordance with the Declaration of Helsinki. A sub-group of participants (n = 9; age 19.2 ± 0.4 y; body mass 76.8 ± 8.9 kg) undertook an additional exercise testing session, as outlined below. All participants provided written informed consent. An a priori sample size calculation was undertaken using a desired power value of 0.8, an alpha level of *p* = 0.05 and effect size values based on the MFO and Fat_max_ data from Ara et al. [17] and Larsen et al. [18], who previously compared these variables during upper and lower body exercise. The calculation resulted in a recommended sample of n = 6. Therefore, our sample of n = 13 was sufficient to account for any potentially large variation in MFO and Fat_max_ responses and any participant drop-out, as well as being in line with previous studies (n = 10 and n = 7, respectively).

The study was a repeated measure design. Inclusion criteria were: young, healthy males who took part in structured exercise sessions at least three times each week. Exclusion criteria were: being specifically cycle or arm ergometry trained or endurance athletes. Recruitment occurred through existing researcher networks and participant friendship groups. Participants visited the university’s exercise physiology laboratory on four separate occasions. Testing sessions were undertaken in the morning, at the same time of day, with at least three days between sessions. Participants were requested not to consume any food within the two to three hours preceding each testing session and to keep the same diet on all testing days, which was confirmed verbally. Participants were also asked not to undertake any strenuous exercise or consume any alcohol or caffeinated products in the 24 h prior to testing. Participants were able to drink water ab libitum during each testing session. Laboratory conditions were maintained at approximately 19–20 °C and ~40% relative humidity. All participants wore similar attire on each laboratory visit, being: light-weight tracksuit trousers, shorts, sports socks and training shoes. Data were collected by the lead author, with considerable experience in both upper and lower body exercise testing.

### 2.2. Exercise Tests

The testing sessions involved two ACE tests, two incremental CE tests and, for the sub-group of participants, one submaximal ACE test. The incremental exercise tests were maximal aerobic tests to determine $\dot{V}$O_2peak_ and power output (PO_peak_). Lower body exercise tests were undertaken using a cycle ergometer (Monark 814E, Varberg, Sweden), whereas the upper body exercise tests were undertaken on an electronically braked Lode arm crank ergometer (Lode Angio, Groningen, the Netherlands). The first ACE and CE tests were preliminary tests to habituate the participants to each exercise mode and the testing environment. For ACE, the protocol began at 0 W, with increments of 20 W every 2-min [21], whereas the CE protocol began at 35 W and increased by 35 W every 3 min [10]. Power output was progressively increased until volitional exhaustion occurred or when the desired cadence (i.e., 70 rev·min^−1^) could not be maintained for 10 s. The second ACE and CE protocols were individualised protocols based on each participant’s performance in the preliminary tests. Participants initially warmed up for 5 min on the unloaded ergometer (0 and 35 W for ACE and CE, respectively) before beginning the main protocol at 40% of the PO_peak_ achieved in the preliminary tests. After this stage, the exercise intensity increased by 10% of each participant’s initial PO_peak_ every two minutes for ACE and every three minutes for CE until volitional exhaustion, using the same criteria as in the preliminary tests.

As previous studies of upper body exercise routinely used exercise protocols with stage durations of 2 min [10,21,22], it is possible that metabolic responses may not have adapted to the same extent as observed for CE when using 3 min exercise stage durations. Therefore, a sub-group of participants (detailed above) undertook a submaximal ACE protocol comparing 2- and 3-minute exercise responses. The submaximal protocol involved the participants undertaking the same individual exercise intensities for three minutes at each stage, up to a rating of perceived exertion (RPE, Borg Scale; [23]) for local muscular effort (RPE_L_) of 15 (equivalent to ‘Hard’ exercise on the 6–20 Borg scale). After all exercise tests, participants performed a cool-down on the unloaded ergometer (0 and 35 W for ACE and CE, respectively) at a self-selected cadence, typically 30 to 50 rev·min^−1^ for ACE and 50 to 60 rev·min^−1^ for CE, for at least five minutes to aid recovery. Due to individualised exercise protocols being undertaken, only the order of ACE and CE testing modes could be realistically counterbalanced.

### 2.3. Measurements

Heart rate (HR) was recorded using a Polar heart rate monitor (Polar beat, Kempele, Finland) consisting of a telemetry belt worn around the chest and a monitor. Values were recorded during seated rest and in the last 15 s of each exercise stage. Participants also wore a face mask connected to an online expired gas analysis system (Metamax 3b, Biophysik GmbH Leipzig, Germany) for the determination of oxygen uptake ($\dot{V}$O_2_) and carbon dioxide production ($\dot{V}$CO_2_), minute ventilation ($\dot{V}$E) and the respiratory exchange ratio (RER). Carbohydrate and fat oxidation were subsequently calculated according to standard stoichiometric equations [24] where:
$$Carbohydrate\;metabolism=(1.67\times \dot{V}O_{2})-(1.67\times \dot{V}CO_{2})\;g\cdot min^{-1}and$$
$$Fat\;metabolism=(6.21\times \dot{V}CO_{2})-(4.21\times \dot{V}O_{2})\;g\cdot min^{-1}$$

Fat metabolism was plotted against the exercise stage for each participant’s individualised ACE and CE protocols and the exercise intensity (%$\dot{V}$O_2peak_) at which the MFO occurred was determined (i.e., Fat_max_). Ratings of perceived exertion were recorded at the end of each exercise stage for RPE_L_ and central (cardiorespiratory) exertion (RPE_C_). The RPE scale and verbal anchors were fully explained to each participant. Capillary blood samples (20 µL) were taken from the participant’s earlobe at rest, at volitional exhaustion and following 5 min of active recovery for the analysis of blood lactate concentration (Biosen, C_Line analyser, EFK Diagnostics, Cardiff, UK). Blood sampling was undertaken in accordance with the British Association of Sport and Exercise Science guidelines [25].

### 2.4. Data Analysis

Data are expressed as mean ± standard deviation (SD). All data were analysed using the Statistical Package for the Social Sciences (v22; IBM Inc., Chicago, USA). Data were checked for normality using the Shapiro–Wilkes test and the homogeneity of variance using Levene’s test. Peak physiological and perceptual responses, maximal carbohydrate oxidation, MFO and Fat_max_ were compared between exercise modes using paired t-tests. Data for physiological variables at each exercise stage and between exercise modes were analysed using factorial analysis of variance (ANOVA), with repeated measures on both factors (Mode: ACE, CE; × intensity; rest, 40, 50, 60, 70 80, 90, 100% PO_peak_). Where significance was achieved, Tukey’s post-hoc analysis was undertaken to determine the difference required between means for significance at the level of *p* < 0.05 [26].

Due to the nature of the exercise protocols increasing by 10% PO_peak_ at each stage, the data could not be statistically compared at a given power output across exercise modes. Therefore, the data for HR, $\dot{V}$O_2_, carbohydrate oxidation and fat oxidation were also plotted for illustrative purposes only against the mean power output representing each 10% increment without statistical analysis. Data for the 2- and 3-minute comparison during ACE for intensities completed by all participants (i.e., up to 70% PO_peak_) were analysed by factorial analysis of variance (ANOVA), with repeated measures on both factors (intensity; 40, 50, 60, 70% $\dot{V}$O_2peak_; × stage duration; 2-min, 3-min).

## 3. Results

### 3.1. Peak Physiological Responses

The peak physiological responses for ACE and CE are shown in Table 1. Peak $\dot{V}$O_2_, $\dot{V}$E, power output and heart rate (HR_peak_) were lower during ACE than for CE (*p* < 0.05). No differences were observed between modes for peak RER, blood lactate concentration or RPE_L_, whereas RPE_C_ approached significance (*p* = 0.065), with RPE_C_ tending to be lower during ACE compared to CE. Carbohydrate oxidation at volitional exhaustion was similar for ACE and CE (4.91 ± 0.73 and 5.16 ± 1.88 g·min^−1^ for ACE and CE, respectively) (*p* = 0.661). Fat oxidation at the end of the exercise was also similar between trials (ACE: 0.06 ± 0.42 g·min^−1^; CE: 0.13 ± 0.67 g·min^−1^; *p* = 0.781).

### 3.2. Submaximal Exercise Responses

Heart rate, $\dot{V}$O_2_ and RER responses during incremental ACE and CE are shown in Figure 1. When expressed relative to peak power (i.e., %PO_peak_), $\dot{V}$O_2_ was lower during ACE than for CE at every exercise intensity (*p* < 0.05). No mode × intensity interactions were observed for HR (*p* = 0.229), RER (*p* = 0.498), RPE_C_ (*p* = 0.892) or RPE_L_ (*p* = 0.969) when expressed relative to %PO_peak_. Main effects for intensity were observed for HR, RPE_C_, RPE_L_ and RER (*p* < 0.05), with additional main effects for exercise modes observed for HR and RPE_C_ (*p* < 0.05). When plotted against absolute power output, $\dot{V}$O_2_ for ACE and CE was similar where the power output overlapped (~90 to 140 W), whereas HR, RPE_C_, RPE_L_ and RER demonstrated greater values during ACE.

### 3.3. Substrate Utilisation

The resting respiratory exchange ratio was similar between trials (0.78 ± 0.11 vs. 0.78 ± 0.06 for ACE and CE, respectively; *p* = 0.978), as were carbohydrates (0.16 ± 0.28 vs. 0.17 ± 0.36 g·min^−1^, respectively) and fat oxidation (0.17 ± 0.07 vs. 0.16 ± 0.06 g·min^−1^, respectively). No mode × intensity interaction was observed for carbohydrate oxidation (*p* = 0.924), although values increased with exercise intensity (intensity main effect, *p* < 0.01). When plotted as absolute exercise intensities, carbohydrate oxidation was greater during ACE (Figure 1).

A significant mode × intensity interaction was observed for fat oxidation (*p* = 0.013), with differences observed between modes for each exercise intensity (*p* < 0.01). Maximal fat oxidation for ACE and CE modes is shown in Figure 2. Maximal fat oxidation was lower in ACE (0.44 ± 0.24 g·min^−1^; range: 0.11–0.79 g·min^−1^) when compared to CE (0.77 ± 0.31 g·min^−1^; range: 0.33–1.23 g·min^−1^; *p* < 0.01), occurring at 53 ± 21 and 67 ± 18%$\dot{V}$O_2peak_, respectively (*p* < 0.01). 

The relationship between resting fat oxidation and MFO for both modes of exercise combined was significant (R = 0.505; R^2^ = 0.254, *p* = 0.020), whereas when considered for ACE and CE separately, only CE elicited a significant relationship (ACE: R = 0.520; R^2^ = 0.270, *p* = 0.151; CE: R = 0.661; R^2^ = 0.436, *p* = 0.019). Based on the coefficient of variation, the inter-participant variability for MFO was greater during ACE (CV = 54.5%) when compared to CE (CV = 40.3%). However, inter-individual variation for MFO was considerably greater than for peak physiological responses such as $\dot{V}$O_2peak_ during ACE and CE (12.7% and 23.4%, respectively).

### 3.4. Comparison of 2- and 3-Minute Exercise Durations

The physiological responses to submaximal ACE during 2- and 3-minute exercise stages are shown in Table 2. No stage duration × intensity interactions were observed for HR, $\dot{V}$O_2_, $\dot{V}$CO_2_, RER, RPE_C_, RPE_L_, carbohydrate oxidation or fat oxidation during the submaximal ACE trials (all *p* > 0.05; Table 2). However, with the exception of fat oxidation, all physiological and perceptual responses increased with exercise intensity (main effects for intensity; all *p* < 0.05).

## 4. Discussion

This is the first study to evaluate the magnitude of MFO and Fat_max_ during individualised incremental exercise tests for upper and lower body exercise in young, recreationally active males. The magnitudes of both MFO and Fat_max_ were lower during ACE than CE but greater than those for previous studies involving overnight fasts. The variability of MFO and Fat_max_ for ACE was greater than for CE.

### 4.1. The Peak Physiological Responses

The peak physiological responses to incremental upper and lower body exercise were indicative of those expected for the population studied, both from our laboratory and others [10,12,22,27,28,29]. In brief, $\dot{V}$O_2peak_ and PO_peak_ for ACE were approximately 75% and 55% of the values observed for CE, respectively, and within the expected range [10,13,27]. Lower peak physiological values for ACE, when compared to CE, are a result of the smaller muscle mass and maximal cardiac output and a greater local fatigue component during upper body exercise [12,30,31,32]. As such, these factors prevent the central cardiorespiratory responses from eliciting the greater values observed during lower body exercise. The peak physiological responses were, therefore, as expected for the population studied.

### 4.2. Incremental Exercise Responses

During incremental exercise, $\dot{V}$O_2_ was lower during ACE than CE when expressed in relation to %PO_peak_. The lower $\dot{V}$O_2_ at each stage of the individualised incremental exercise tests is a result of the lower $\dot{V}$O_2peak_ during ACE and likely systematic. Indeed, the difference between $\dot{V}$O_2peak_ values for ACE and CE was similar to the difference between modes for each individualised exercise stage (~1.0 L·min^−1^). Conversely, HR was greater during ACE than CE at each individualised stage, although representing a similar systematic exercise mode difference (overall mean between modes across exercise intensities ~8 beats·min^−1^). When plotted in relation to absolute power output, $\dot{V}$O_2_ was similar at the exercise intensities accomplished during both upper and lower body exercise (~90 to 140 W), reflecting the overall energy requirements for a given power output [11]. 

During ACE at a given power output, HR, RER, carbohydrate metabolism and RPE_L_ were greater and fat oxidation was lower when compared to CE. Such responses illustrate the greater physiological strain at a given power output when exercising with the smaller muscle mass of the arms during ACE when compared to the legs during CE [10,12,27,28]. Subsequently, the lower RPE_C_ during both individualised and absolute power outputs for ACE is a result of the resultant peripheral limitation during this exercise mode [10]. The physiological and perceptual responses to incremental ACE and CE exercise are, thus, in line with previous research.

### 4.3. Maximal Fat Oxidation during CE

The MFO values observed for CE in the current study (~0.77 g·min^−1^) are greater than those reported for previous studies of CE and treadmill exercise (0.43 and 0.65 g·min^−1^, respectively; [14]) and those reported in a recent review of MFO (endurance-trained males ~0.53 g·min^−1^; recreationally active males ~0.43 g·min^−1^; [7]). However, the mean MFO values are well within the wide range of values reported for males in the literature (i.e., 0.17 to 1.27 g·min^−1^; mean = 0.60 g·min^−1^ [6,33,34,35]). The greater MFO values in the current study may be a result of the resting metabolic state of the participants. Resting RER values (~0.78) were indicative of a predominance of fat metabolism, also evidenced by resting fat oxidation values (~0.17 g·min^−1^) being greater than those previously reported (0.06 g·min^−1^; [36]). Rosenkilde et al. [37] observed that participants with lower resting RER values (0.76) elicited greater MFO than those with greater resting RER values (0.86). Furthermore, Robinson et al. [33] observed a relationship between 24-hour fat oxidation and MFO, suggesting habitual diet may be an important contributor to the maximal rate of fat oxidation during exercise.

The current study demonstrated a correlation between resting fat oxidation and MFO values, confirming that greater resting fat oxidation resulted in greater maximal fat oxidation. Thus, although participants in the current study were asked not to eat for only 2–3 h prior to exercise, it is possible that their habitual diets affected their resting and subsequent exercising metabolic substrates. However, as most individuals are unlikely to exercise after an overnight fast, the current MFO and Fat_max_ values are greater than expected for fasted conditions. The values are still within the expected range of reported values reflecting true maximal fat oxidation rates in daily, ecologically valid conditions.

### 4.4. Maximal Fat Oxidation during ACE

Maximal fat oxidation rates and Fat_max_ were lower during ACE than CE and are in agreement with the current hypothesis. Few studies have reported Fat_max_ during upper body exercise, except for those that have reported values for control groups matched to diabetic or obese experimental groups [17,18] or those with spinal cord injury [19]. For example, Ara et al. [17] reported lower MFO values for incremental ACE than for CE (0.18 vs. 0.28 g·min^−1^) in previously obese men (31 y, $\dot{V}$O_2peak_ during CE: 27 mL·kg·^−1^min^−1^). Larsen et al. [18] also reported lower MFO values in ACE than CE (0.18 vs. 0.30 g·min^−1^) in age- and body-mass-matched controls (43 y, $\dot{V}$O_2peak_ during CE: 42 mL·kg·^−1^min^−1^). Both studies reported Fat_max_ at relatively low percentages of $\dot{V}$O_2peak_ in both ACE and CE trials (~30 vs. ~40%$\dot{V}$O_2peak_, respectively). However, the exercise intensity at which MFO occurred and the absolute MFO values represent similar ACE:CE ratios to the current study (i.e., 0.74 and 0.75, 0.79 and 0.64, and 0.60 and 0.57 for Ara et al. [17], Larsen et al. [18], and the current study, respectively).

Jacobs et al. [19] observed greater MFO values during ACE in sedentary participants with spinal cord injury compared to a non-disabled control group of men and women, suggesting that the greater training status of the upper body through habitual wheelchair locomotion may have adapted upper body fat metabolism. However, MFO and Fat_max_ values for both groups were still low (0.07 to 0.13 g·min^−1^ at 42%$\dot{V}$O_2peak_ and 0.02 to 0.06 g·min^−1^ at 14%$\dot{V}$O_2peak_, respectively). The greater overall training status of the current participants, based on $\dot{V}$O_2peak_ values during ACE and CE (and no sympathetic nervous system dysfunction), would likely have contributed to the greater absolute MFO values observed. Thus, although our current understanding of MFO during ACE is from older and less aerobically conditioned individuals than in the present study, values across exercise modes appear to occur at similar proportions of maximal capacity and with a large potential for improvement in less active individuals. 

### 4.5. Inter-Individual Variability

Usually, only small amounts of variation can be accounted for when examining factors that influence MFO and Fat_max_. As with previous studies, the current study demonstrates a significant positive correlation between MFO and $\dot{V}$O_2peak_ across both modes of exercise; however, the R^2^ indicated that only 25% of the variation in MFO could be accounted for. Although lower than the 35% and 47% of variation accounted for in MFO for more than one contributing component by Randell et al. [33] and Venables et al. [6], respectively, our R^2^ value is similar in that much of the variation cannot be accounted for. Key factors reported to be related to MFO include maximal $\dot{V}$O_2_, body fat percentage, fat-free mass, fast duration, self-reported physical activity and sex [6,33]. Randell et al. [33] recently indicated that genetic or epigenetic factors may account for a large proportion of the unexplained variance and should be explored further.

The inter-individual variation for MFO in the current study, as indicated by the coefficient of variation, was extremely broad, with ACE being greater than CE (54.5 vs. 40.3%, respectively) and much greater than for $\dot{V}$O_2peak_ per se (23.4 vs. 12.7%, respectively). When compared to CE, the variability of MFO may be further confounded during upper body exercise when considering individual fibre type differences and the greater range of training statuses. 

### 4.6. Application

The main application of the current data is in determining potential aerobic exercise training intensities for optimising fat metabolism and enabling weight loss. A small number of studies have examined training regimes utilising Fat_max_ exercise intensity [38,39,40,41]. These training studies have predominantly involved overweight or obese participants, subsequently focusing upon a range of health-related outcome measures: observing improvements in body composition, cardiovascular function, strength and flexibility [39,40]. Within these studies, Fat_max_ ranged between 44–54%$\dot{V}$O_2max_ [38,39,40] for lower body exercise, which is lower than for the current young, recreationally active participants during CE (67%$\dot{V}$O_2peak_), most likely due to their aerobic fitness status. Although upper body Fat_max_ intensities for obese, previously obese and age- and mass-matched control participants are lower than for lower body exercise [17], the use of Fat_max_ as an upper body training intensity has not been reported.

Using a traditional aerobic exercise training intensity (60% PO_peak_) in a similar population to the current study, Bottoms [42] observed increased free fatty acid concentrations at the end of 30 min continuous ACE, following 8 weeks of aerobic ACE training. The equivalent exercise intensity post-training was ~57%$\dot{V}$O_2peak_ and similar to that for ACE Fat_max_ in the current study (53%$\dot{V}$O_2peak_). Furthermore, Bottoms [42] utilised two continuous training sessions and one interval training session per week. In light of the finding that continuous low-intensity exercise training sessions can improve fat metabolism when compared to interval-type training, the ratio of 2:1 for continuous to interval exercise training sessions appears potentially useful for improving fat metabolism in young, healthy, recreationally active male participants. The continuous- and interval-type exercise training comparison noted above, however, was undertaken in overweight men, so the finding needs to be confirmed for healthy individuals.

### 4.7. Limitations

Firstly, this study examined MFO and Fat_max_ values in male participants only. Previous studies of trained males and females during prolonged exercise at different intensities (25, 65, 85%$\dot{V}$O_2max;_) observed similar total fat oxidation responses, with the greatest values occurring at 65%$\dot{V}$O_2max_ [43,44]. More recently, similar MFOs in young, trained men and women, despite major differences in plasma lipid concentrations during graded exercise, were also observed [45]. 

In contrast, a review comparing MFO and Fat_max_ between men and women reported MFO to be greater in males and Fat_max_ to be greater in females [7], though their study did not take into account the separate effects of age, training status, mode of exercise and weight status [46]. Although there are no reported studies on fat metabolism during incremental ACE in females, it is clear that peak physiological responses during ACE in females are lower, but with a similar ratio between ACE and CE, with $\dot{V}$O_2peak_ for ACE being ~63% [47] to 68% of CE values [48]. It has also been reported that females are 52% and 66% as strong as their male counterparts in the upper body (elbow flexors) and lower body (knee flexors), respectively, which is due to larger fibre areas in males [49]. Although differences in the strength and aerobic power of females vary to a greater extent between the upper and lower body compared to men, there are no data to suggest sex differences in MFO and Fat_max_ for the upper body. A similar response in MFO and Fat_max_ may occur for upper body exercise, as observed for males, although this remains to be tested, particularly in consideration of the effects of sex hormones observed on metabolism [50].

Secondly, the current study comprised a heterogenous group with respect to training status, more so for upper body exercise capacity. Thus, examining homogenous upper body trained groups or potential changes in fat oxidation before and after upper body aerobic training may be more enlightening. 

Thirdly, although resting fat oxidation and RER values were similar between ACE and CE trials, habitual dietary habits were not recorded, which may help explain why the MFO values reported were at the higher end of the expected physiological range.

Finally, fat oxidation values at volitional exhaustion had not reached zero, with participants potentially not having reached true volitional exhaustion. However, with the exception of RER_peak_ values, typical end criteria for ACE and CE exercise protocols were achieved, with values in line with previous studies. Furthermore, at volitional exhaustion, fat oxidation had returned to similar values as those observed at baseline. If greater peak exercise capacity had been achieved, it is likely that this would only have been one, or part of one, further exercise stage that would not affect the MFO values per se and would only slightly reduce the relative exercise intensity at which it occurred (i.e., Fat_max_). We are, therefore, confident we can state that both MFO and Fat_max_ during ACE are lower than for CE and demonstrate greater variability.

### 4.8. Future Research

The following areas for future research are recommended. The short duration pre-exercise fast and non-specific dietary regime of the current study likely contributed to the relationship between resting fat oxidation and MFO, prompting the suggestion that habitual diet composition should be considered. Future examination of MFO during exercise, following participants’ habitual diets and standardised diets, as well as ecologically valid fast durations, will aid our understanding of dietary factors affecting MFO. The potential to maximise MFO during both ACE and CE will also be enhanced with this approach, along with our understanding of the individual variation in MFO during both upper and lower body exercise. 

Future studies on MFO and Fat_max_ adaptations during upper body exercise training in otherwise healthy individuals will provide an adjunct to traditional training modes from which additional physiological and functional benefits (e.g., activities of daily living) may be achieved. Similarly, the application of upper body exercise training to clinical populations allows the potential for improving fat metabolism in populations either without the use or with limited use of their legs. 

Further studies to examine the potential underlying genetic and epigenetic factors across a range of exercise modes and populations are also recommended to help understand the intra- and inter-individual variability of MFO and Fat_max_.

## 5. Conclusions

The results of this study show consistently lower MFO and Fat_max_ during upper body exercise in young, healthy, recreationally active male participants compared to lower body exercise. Although inter-individual variation was greater for MFO and Fat_max_ during upper body exercise, these data have important applications for weight loss strategies in healthy male populations and those unable to use their legs or where upper body exercise may be the only available exercise mode.

## Figures and Tables

**Figure 1 ijerph-19-15311-f001:**
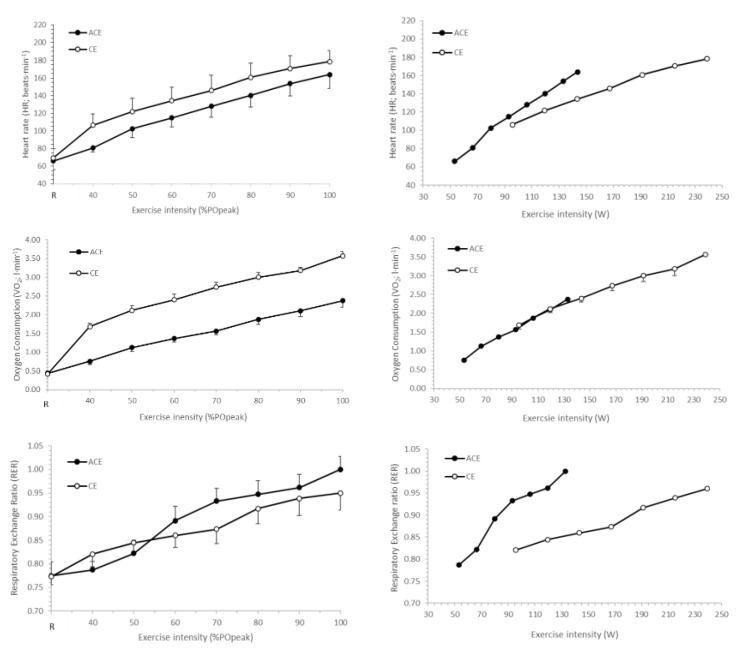
Heart rate (HR), oxygen uptake ($\dot{V}$O_2_) and respiratory exchange ratio (RER) responses to incremental arm crank (ACE) and cycle ergometry (CE). Data plotted in relation to %PO_peak_ (**left** panel) and absolute power output (W) during exercise (**right** panel).

**Figure 2 ijerph-19-15311-f002:**
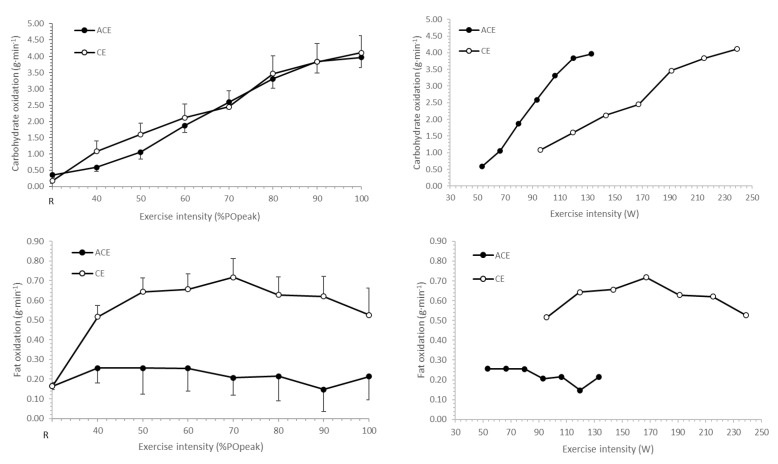
Carbohydrate and fat oxidation responses to incremental arm crank (ACE) and cycle ergometry (CE). Data plotted in relation to %PO_peak_ (**left** panel) and absolute power output (W) during exercise (**right** panel). NB: Of the n = 13 participants, only n = 9 completed the final exercise stage at 100%PO_peak_; therefore, data for ACE at 100% POpeak and absolute power output of 133 W are for the remaining n = 9.

**Table 1 ijerph-19-15311-t001:** Peak physiological responses for arm crank ergometry (ACE) and cycle ergometry (CE) in young, healthy males.

	ACE	CE	*p*
	Mean ± SD	Mean ± SD	
$\dot{V}$O_2 peak_ (L·min^−1^)	2.73 ± 0.64	3.62 ± 0.46	<0.001
$\dot{V}$O_2peak_ (mL·kg·^−1^min^−1^)	36 ± 7	48 ± 5	<0.001
$\dot{V}$E_peak_ (L·min^−1^)	100.4 ± 19.6	120.1 ± 19.4	0.003
PO_peak_ (W)	133 ± 21	239 ± 33	<0.001
HR_peak_ (beats·min^−1^)	170 ± 16	182 ± 11	0.020
RPE_C_ (Borg Scale)	16 ± 3	18 ± 2	0.065
RPE_L_ (Borg Scale)	19 ± 1	19 ± 1	0.120
BLa_peak_ (mmol·L^−1^)	7.6 ± 1.9	9.0 ± 2.9	0.134
BLa+5 (mmol·L^−1^)	8.0 ± 1.7	8.4 ± 3.0	0.179
RER_peak_	1.00 ± 0.10	0.96 ± 0.13	0.328

Note: $\dot{V}$ O_2peak_ (L·min^−1^) = absolute peak oxygen uptake; $\dot{V}$ O_2peak_ (mL·kg·^−1^min^−1^) = relative peak oxygen uptake; $\dot{V}$ E_peak_ (L·min^−1^) = peak minute ventilation; PO_peak_ (W) = peak power output; HR_peak_ (beats·min^−1^) = peak heart rate; RPE_C_ (Borg Scale) = peak rating of perceived exertion (central/cardiorespiratory effort); RPE_L_ (Borg Scale) = peak rating of perceived exertion (local muscular effort); BLa_peak_ (mmol·L^−1^) = peak blood lactate concentration, BLa+5 (mmol·L^−1^) = blood lactate concentration 5-min post-cessation of exercise; RER_peak_ = peak respiratory exchange ratio.

**Table 2 ijerph-19-15311-t002:** Physiological responses (mean ± SD) for submaximal intensities (%PO_peak_) of arm crank ergometry (ACE) during 2- and 3-minute exercise stages in young, healthy males (n = 9).

	Stage		%PO_peak_				*p*	
		40	50	60	70	%PO_peak_	Stage	Int.
HR	2 min	103 ± 22	122 ± 19	135 ± 22	151 ± 22	<0.05	0.556	0.997
(beats·min^−1^)	3 min	107 ± 23	125 ± 19	138 ± 22	152 ± 23			
$\dot{V}$O_2_	2 min	0.65 ± 0.11	1.04 ± 0.30	1.34 ± 0.31	1.58 ± 0.33	<0.05	0.983	0.991
(L·min^−1^)	3 min	0.62 ± 0.11	1.07 ± 0.35	1.34 ± 0.33	1.58 ± 0.35			
$\dot{V}$CO_2_	2 min	0.57 ± 0.16	0.94 ± 0.21	1.23 ± 0.25	1.43 ± 0.31	<0.05	0.773	0.919
(L·min^−1^)	3 min	0.57 ± 0.15	1.03 ± 0.22	1.23 ± 0.26	1.42 ± 0.41			
RER	2 min	0.83 ± 0.03	0.88 ± 0.03	0.95 ± 0.03	0.96 ± 0.03	<0.05	0.145	0.774
	3-min	0.83 ± 0.03	0.92 ± 0.03	0.96 ± 0.04	0.95 ± 0.04			
FOx	2 min	0.16 ± 0.04	0.20 ± 0.06	0.22 ± 0.08	0.31 ± 0.14	0.324	0.963	0.986
(g·min^−1^)	3 min	0.16 ± 0.04	0.17 ± 0.06	0.25 ± 0.10	0.33 ± 0.15			
CHO	2 min	0.81 ± 0.78	1.49 ± 0.63	2.00 ± 0.82	2.20 ± 1.57	0.003	0.692	0.918
(g·min^−1^)	3 min	0.92 ± 0.96	1.88 ± 0.83	1.95 ± 1.02	2.15 ± 1.62			
RPE_C_	2 min	7.2 ± 0.7	8.8 ± 1.0	10.6 ± 1.3	12.6 ± 1.1	<0.05	0.145	0.774
(Borg Scale)	3 min	7.2 ± 0.7	9.1 ± 1.1	10.2 ± 1.3	13.0 ± 1.6			
RPE_L_	2 min	8.4 ± 1.2	11.0 ± 1.2)	13.1 ± 0.8	14.3 ± 0.9	<0.05	0.088	0.968
(Borg Scale)	3 min	8.9 ± 1.1	12.0 ± 1.0	14.3 ± 0.8	14.8 ± 0.8			

Note: HR (beats·min^−1^) = heart rate; $\dot{V}$ O_2_ (L·min^−1^) = absolute oxygen uptake; $\dot{V}$ CO_2_ (L·min^−1^) = absolute carbon dioxide production; RER = respiratory exchange ratio; FOx (g·min^−1^) = fat oxidation; CHO (g·min^−1^) = carbohydrate oxidation; RPE_C_ (Borg Scale) = rating of perceived exertion (central/cardiorespiratory effort); RPE_L_ (Borg Scale) = rating of perceived exertion (local muscular effort); p = p-value; %PO_peak_ = exercise intensity main effect (40 vs. 50 vs. 60 vs. 70% PO_peak_); Stage = stage duration main effect (2-min vs. 3-min), Int = %PO_peak_ × Stage interaction.

## Data Availability

The data presented in this study are available on request from the corresponding author.

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
