# Peer review of "Maximal Fat Oxidation during Incremental Upper and Lower Body Exercise in Healthy Young Males"

_ijerph, 2022, doi:10.3390/ijerph192215311_

Round 1

Reviewer 1 Report

It was with great enthusiasm that I volunteered to review this manuscript. The manuscript sought to compared upper versus lower body on fat oxidation. The results of the study demonstrate that upper body exercise elicits a lower fat oxidation compared to lower body. The study has a strong hypothesis, and quality methodology. I do have a few things that may strengthen this manuscript and have highlighted them below. 

-Keywords: if keywords are in the title then adding them here is duplicative.

-Intro: The way it is currently written, the authors state, 'when maximal fat oxidation occurs has been termed 'fat max'.' But from there the authors use these as two different entities. I can't tell if the terms are the same or not, and was continually confused. 

-Order is important. In 2.2 the order shifts, from ACE and CE, to CE and ACE. Maintaining order assists readability. Please make sure that is ACE is presented first to CE throughout the document. I noted in figure 1 it's CE then ACE in the legend, but is ACE and then CE in the caption.

-Page 3, line 101. 'warmed down', this is not something I have seen previously. 'Cool down' maybe?

- TAble 2. This needs to be spaced better, as some data were on page 6 some on page 7, for the same variable. 

Reviewer 2 Report

The major issue I found is that there was no diet control or, at least, diet analysis prior to the test. Despite the test result solely relying on the gas exchange at the external respiration, the influence from the macronutrients of "the habitual diets" is not considered and controlled.

Reviewer 3 Report

Dear authors,

I would like to express my gratitude regarding the opportunity to review this manuscript.

The study topic is interesting, at this stage the manuscript requires many improvements. Below comments and suggestions with line indication:

22 – “respectively P<0.01” – Please review the text.

22 – Please change “.” in the units by middle dot. This should be applicable in all manuscript.

24-25 – “MFO was higher in CE but VO2peak was also higher”. Please consider rephrasing in these lines.

33 – No space between citation numbers in this line, and please review all manuscript regarding this detail.

36 – In the last consideration, “and” is suggested.

30-60 – Please develop the introduction section, namely with more literature regarding the study topic. It is also advisable to consider more paragraphs, with smaller number of lines, to improve the conditions for readers.

65 – Reference to Helsinki declaration is suggested.

68 – Please provide information related to informed consents.

74 – Please describe the procedures in detail. For example, where were the evaluations performed? Conditions (temperature, humidity?) Time of day (circadian effect). Previous nutrition? Clothes during data collection? Who collected the data, training, and experience? All details should be considered and detailed.

78, 111, 112 – Please correct the symbol text. Please review all manuscript considering this indication.

84 and 85 – “20 W” and “35W”, please standardize. Same in page 89 (“35W and 0 Watts)”.

94-96 – Please indicate the studies references.

101 – “(35Wand 0W,” Please correct.

87 – “sec” / 109 “15 s” – Please Review in these lines and carefully review all manuscript considering standardization of these and other examples.

118 – MFO previously abbreviated.

119 – “RPE” suggested.

127 – Please review upper and lowercase.

127-144 – Please consider paragraphs. Please include sample power details.

149 – Please improve the English “were no different between modes”.

153 – “0.13 ±0.67” / “0.06 ± 0.42” – Please review space between mean and SD in this line and throughout the manuscript (for example in the abstract this same detail is present).

155 – Please review and implement journal template and instructions for authors. The title format is not accordingly.

156 – Please format the table content. Some examples: Some text is bold, other not, same number of decimals in the values, borg scale number indication and in the first column, text aligned.

157 – Please review the space.

159 – “Heart rate, V ̇O2” please correct.

168 – Please introduce figure 1 with text.

169 - Please review and implement journal template and instructions for authors. The legend format is not accordingly.

169 – Please improve the figure quality.

180 – Paragraph suggested.

181 – Please review “-1” below and above line.

183 – “MFO”.

186 – Paragraph suggested.

190 – Please introduce figure 2 with text. Please review and implement journal template and instructions for authors. Please improve the figure quality.

200 - Please review and implement journal template and instructions for authors. The title format is not accordingly.

200 – Indication of n=9, the abstract indicates n=13 (lines 63 and 97 the same). This may not be clear for readers. Please consider precise information.

201 – “-1“ above text.

201 – Please reformulate table content aiming interpretation by readers.

202 – Legend suggested (e.g. “PMP”).

221-238 / 240-261 / 263-286 / 288-303 / 305-327 / 329-356 / 364 – Please review the text and consider paragraphs to improve reading conditions.

373 – Please consider indicating suggestions for future research.

376 – “male” is missing.

375-382 – Please consider highlighting the main findings of the study associated to take-home messages, and desirably, with practical applications.

399 – All references format should be corrected considering the journal template and instructions for authors.

Please carefully review all manuscript and consider English improvement.

Round 2

Reviewer 2 Report

The authors made great effort for extensive revision was done in short time.

Author Response

See attached file - NB: this is to thank the reviewer as not further comments were raised.

Reviewer 3 Report

Dear authors,

Thank you for considering my suggestions and incorporating them into the manuscript. 

Below suggestions related to this last version (v2), with line indication.

23 – “53 ±21” - Please review space between text. This should be considered throughout the manuscript.

29 forward – Please justify all text.

55-56 / 68-69 – Please correct the paragraph space, the text should be continuous. Please see the journal instructions for authors and consider all manuscript regarding this detail.

34 – First time “maximal or peak oxygen uptake”, not in lines 44-45. Please review the abbreviations in all manuscript.

49 & 50 – Please review the unit format - ml.kg.-1min.-1 not only in these pages, but throughout the manuscript.

96 – “maximal fat oxidation” – Please abbreviate.

96 & 97 – “et al.” – point is missing. Please review throughout the manuscript.

172 – Please insert SPSS City and Country.

183 – First time in the text “HR” should be in full previously to abbreviation.

203 – Please format the table content and place in the manuscript according to the journal template.

203 – Please standardize the units of the values in the table. For example, relative ?̇O2peak should presented decimals.

204 – A legend should be in the end of the table describing the variables.

215 – Figure 1 y-axys does not present middle dot´s, please review. The same in figure 2.

219-221 / 299-230 – Please review “0.78 ±0.11vs” – spaces in the values. This should be considered in all manuscript.

227 – “. Maximal” more than one space, please correct.

232 – I believe “MFO”, please review.

240 – Please delete.

244 – Please standardize throughout the manuscript “133W”

276 – “beats.min-1”, please change to middle dot and review all manuscript regarding this detail.

309 – “et al” – End point. Please review in all text (for example the same again in 329 & 335).

332 – “~40%?̇O2peak” – Please review.

324-347 – Please consider splitting the paragraph.

392-437 – Please consider reformulating the text aiming lower number of lines and presenting more clear and direct messages.

439-462 - Please consider reformulating the text aiming lower number of lines and presenting more clear and direct messages.

464-469 – The conclusions section can be developed and improved.

480 – End point is missing.

488 - All references format should be corrected considering the journal template and instructions for authors.

Please carefully review all manuscript and consider English improvement before v3.

Round 3

Reviewer 3 Report

Dear authors,

Thank you for considering my suggestions and incorporating them into the manuscript. 

Below suggestions related to this last version (v2), with line indication.

3 – Please remove end point in the title.

204-205 – “Where SPSS returned p values of ‘0.000’ they are reported as ‘p < 0.001’”. The same in 276-277. It is suggested to remove this sentences.

216 – “uptake” instead of “consumption” suggested.

390-446 – Text should be justified.

472 – References format are incorrect. Please pay special attention to this.

Please review:

https://www.mdpi.com/journal/ijerph/instructions

Journal Articles:
1. Author 1, A.B.; Author 2, C.D. Title of the article.
Abbreviated Journal Name Year, Volume, page range.

It is also suggested to monitor recent published articles in the journal.

Briefly, changes are needed considering year in bold without brackets, volume in italic, whenever possible doi (without hyperlink).

A careful review/reading after considering this round 2 review is suggested, with special attention to details and English.
